# An Efficient Method for Vault Nanoparticle Conjugation with Finely Adjustable Amounts of Antibodies and Small Molecules

**DOI:** 10.3390/ijms25126629

**Published:** 2024-06-16

**Authors:** Giulia Tomaino, Camilla Pantaleoni, Annalisa D’Urzo, Carlo Santambrogio, Filippo Testa, Matilde Ciprandi, Davide Cotugno, Gianni Frascotti, Marco Vanoni, Paolo Tortora

**Affiliations:** 1Department of Biotechnology and Biosciences, University of Milano-Bicocca, 20126 Milan, Italy; giulia.tomaino@unimib.it (G.T.); c.pantaleoni@campus.unimib.it (C.P.); annalisa.durzo@unimib.it (A.D.); carlo.santambrogio@unimib.it (C.S.); f.testa@campus.unimib.it (F.T.); matilde.ciprandi@unimib.it (M.C.); davide.cotugno@ieo.it (D.C.); marco.vanoni@unimib.it (M.V.); 2ISBE-SYSBIO Centre for Systems Biology, 20126 Milan, Italy

**Keywords:** vault nanoparticle, major vault protein, *Komagataella phaffii* expression system, antibody-mediated targeting, Fc antibody portion, surface plasmon resonance, liquid chromatography/mass spectrometry

## Abstract

Vaults are eukaryotic ribonucleoproteins consisting of 78 copies of the major vault protein (MVP), which assemble into a nanoparticle with an about 60 nm volume-based size, enclosing other proteins and RNAs. Regardless of their physiological role(s), vaults represent ideal, natural hollow nanoparticles, which are produced by the assembly of the sole MVP. Here, we have expressed in *Komagataella phaffi* and purified an MVP variant carrying a C-terminal Z peptide (vault-Z), which can tightly bind an antibody’s Fc portion, in view of targeted delivery. Via surface plasmon resonance analysis, we could determine a 2.5 nM affinity to the monoclonal antibody Trastuzumab (Tz)/vault-Z 1:1 interaction. Then, we characterized the in-solution interaction via co-incubation, ultracentrifugation, and analysis of the pelleted proteins. This showed virtually irreversible binding up to an at least 10:1 Tz/vault-Z ratio. As a proof of concept, we labeled the Fc portion of Tz with a fluorophore and conjugated it with the nanoparticle, along with either Tz or Cetuximab, another monoclonal antibody. Thus, we could demonstrate antibody-dependent, selective uptake by the SKBR3 and MDA-MB 231 breast cancer cell lines. These investigations provide a novel, flexible technological platform that significantly extends vault-Z’s applications, in that it can be stably conjugated with finely adjusted amounts of antibodies as well as of other molecules, such as fluorophores, cell-targeting peptides, or drugs, using the Fc portion as a scaffold.

## 1. Introduction

Recent years have witnessed a growing impact of nanotechnology in biomedicine due to the remarkable therapeutic and diagnostic potential of nanoparticles (NPs; for extensive reviews, see [1,2]). NPs are frequently employed for cancer treatment, being loaded for this purpose with specific anticancer drugs [3,4] and equipped with monoclonal antibodies (MoAbs), proteins, or peptides that selectively direct them to cancer cells [1,5,6,7]. MoAbs are capable of binding molecular receptors overexpressed at the cancer cell surface, which ensures a targeted delivery and endocytic uptake of the therapeutic agent [8].

Several methods for NP functionalization have been developed, including passive adsorption, covalent binding based on different protocols, including carbodiimide, maleimide, or click chemistry, or binding via adapter molecules. However, these methods often suffer from major disadvantages in that they are complex and time-consuming, and/or do not necessarily guarantee the correct orientation required for MoAb to effectively bind to the target molecule [7].

With regard to the nature of nano-sized materials available to date, there is a wide and diversified repertoire encompassing several varieties of NPs, including inorganic, polymeric, lipidic, and protein-based nanomaterials [1,2]. In particular, in the last decade, the latter have gained growing interest as drug delivery systems, due to the several advantages they offer, especially in terms of a lack of toxicity and low immunogenicity, biodegradability, biocompatibility, size homogeneity, and colloidal stability.

In this context, the vault NP stands out for its unique properties. Vaults are natural ribonucleoproteins found in several eukaryotes [9,10]. In their molecular assembly, the 99 kDa major vault protein (MVP) is present in 78 copies and generates a barrel-like, roughly ovoidal natural NP consisting of two symmetrical halves, whose C-termini form two protruding caps at both ends [11,12]. Vaults also enclose other molecular components, i.e., the 193 kDa vault poly(ADP-ribose) polymerase, the 290 kDa telomerase-associated protein-1 (TEP1), and one or more small untranslated RNAs [13,14,15,16,17]. Overall, the molecular mass of vault particles amounts to about 13 MDa, their size is 72.5 nm × 41 nm × 41 nm, and the internal cavity volume is 5 × 10^4^ nm^3^. Notably, a well-assembled vault structure can be produced by expressing the sole MVP, as originally shown in insect cells [18,19]. Although the physiological roles of this nanocomplex are only partially understood, numerous reports highlight its involvement in several pro-survival functions [10,20].

Thanks to the aforementioned characteristics, this macromolecular assembly has attracted growing attention as a tool for drug/gene delivery, often directed to cancer cells [21]. Actually, it was shown that the sole MVP can assemble into an “empty” vault, which not only can accommodate large amounts of cargo molecules but can also be targeted to specific cell surface receptors, provided it is bound to a targeting molecule (e.g., MoAbs) via genetic [22] or chemical approaches [23]. Of remarkable interest, in this respect, is the development of a vault variant carrying, at the MVP C-terminus, a staphylococcal, protein A-derived sequence, referred to as Z domain or peptide (vault-Z and MVP-Z, respectively), 33 residues in length, which has been reported to tightly bind the Fc portion of human IgG1 [22,24]. The resulting recombinant vaults expose these domains at both caps, thus enabling conjugation with any desired MoAb [22]. Therefore, besides tight antibody binding, these molecular assemblies offer the remarkable advantage of ensuring the proper MoAb orientation, whereby the antigen-binding sites are fully available to the interaction with the cognate antigen, thus circumventing one major drawback associated with many other conjugation strategies [7].

With a view to exploring the vault’s potential as a platform to be variously harnessed to optimize drug delivery, we formerly expressed authentic human vault (i.e., devoid of the Z peptide) in the methylotropic yeast *Komagataella phaffii* (formerly *Pichia pastoris*) [25] following a previously described protocol [26]. Then, we purified it by RNase pretreatment of cell-free extracts followed by size exclusion chromatography (SEC) [25], a procedure representing a modification of the one that we formerly developed starting from transfected insect cell extracts [27].

Here, we have cloned, expressed in *K. phaffi*, and purified the human vault-Z variant following the above protocol, with the initial aim of undertaking an in-depth characterization of its antibody-binding affinity and saturation profile. Thus, we found that up to at least ten MoAb molecules per vault-Z were bound in a virtually irreversible and quantitative fashion, whereas the affinity declined at higher molar ratios. Furthermore, we exploited the Fc domain in isolation as a molecular scaffold for stable conjugation to vault-Z of a fluorescent dye by taking advantage of its high affinity for the Z peptide. Thus, the possibility of binding to the NP, either MoAbs or Fc, or both, establishes a novel technological platform entailing different benefits. First, not only can Fc be conjugated with fluorescent dyes, as we have achieved in the present study but, prospectively, also with a wide repertoire of small therapeutic molecules, such as, for instance, doxorubicin and taxanes [28,29], as well as cell-targeting peptides (most often identified by phage display technology [30]). This substantially expands the scope of the vault-Z-based platform in view of future therapeutic applications. Second, this setup makes it possible to finely tune the molar ratio vault/ligand so as to optimize both fluorophore and antibody conjugation, which might ensure optimized vault delivery to target cells. Third, it also streamlines the conjugation protocols by minimizing vault manipulation, thus circumventing the inherent issues related to its stability and recovery during the chemical process.

As a proof of concept, we conjugated vault-Z NP with fluorophore-bound Fc and either Trastuzumab (Tz) or Cetuximab (CTX), which dock to Her2 and EGFR receptors, respectively. These receptors are overexpressed in several tumors [31,32], which makes them suitable for targeted delivery. In particular, in our assays, Tz and CTX efficiently targeted vault-Z to two different mammary carcinoma cell lines. Overall, our results validated the strategy we have devised for vault delivery optimization.

## 2. Results

### 2.1. The Production and Purification of the Vault-Z Variant

The protocol for the purification of the authentic vault (i.e., devoid of the Z peptide) from *K. phaffi* and in-depth characterization of its purity and properties have been previously described [25]. In particular, the purification procedure relies upon an RNase pretreatment of cell-free extracts, followed by size exclusion chromatography (SEC) using Sepharose CL-6B as the matrix.

Here, we used the same protocol for vault-Z production and purification. SDS-PAGE and Western blot analysis of purified protein confirmed the identity and purity of the protein. Furthermore, TEM and DLS displayed the expected morphology and one, essentially symmetrical peak, respectively (Appendix A).

### 2.2. Investigating Trastuzumab Binding Mode to Either Vault-Z or the Sole Z Domain by Surface Plasmon Resonance Analysis

To determine Trastuzumab’s (Tz) affinity for vault-Z as well as the relevant binding and release rate constants, we performed surface plasmon resonance (SPR) analyses using a Biacore X100 (Cytiva-Pall, Marlborough, MA, USA). In our setup, the antibody was immobilized on a CM3 sensor chip via amine-coupling chemistry at 1220 response units (R.U.). The reverse setup, i.e., vault immobilization, was not feasible, as this procedure must be performed at a pH value of 4.5 (Section 4.8), which is incompatible with the preservation of vault structural integrity. The real-time association and dissociation rates of the binding of vault-Z, authentic vault, and the Z domain in isolation to Tz were recorded. Binding kinetics were reproducible in the different experiments.

As expected, the authentic vault showed virtually complete absence of interaction with the antibody (Figure 1). As regards vault-Z, scalar Tz concentrations in the range of 1 to 60 nM were manually injected in sequence in the multi-cycle kinetics mode (MCK) and binding and release kinetics were recorded (Figure 2).

Sensorgrams were fitted using a Langmuir 1:1 binding model to derive separate k_a_/k_d_ (association/dissociation) constants, in keeping with a simple bi-molecular reaction [33]. From each binding–release profile, k_a_ and k_d_ rate constants, along with the relevant equilibrium dissociation constant (K_D_), were determined. The mean ± standard deviations of the resulting values are presented in Table 1, which highlighted a 2.53 nM vault-Z affinity.

We then determined kinetic and equilibrium binding constants of Tz to the Z peptide in isolation (Figure 3). An over ten-fold lower affinity (38.3 nM) than that of vault-Z was determined (Table 1).

Because of the fast dissociation binding kinetics (k_d_) between Z peptide and Tz, calculated by the fitting binding 1:1 (Table 1), the set of SPR sensorgrams shown in Figure 3 was also fitted with the affinity model by using the same BiacoreX100 BIA Evaluation software version 2.0.2. The resulting affinity fitting with the plot of the calculated Req versus Z peptide concentrations yielded a K_D_ value of 56 nM (Appendix A), in satisfactory agreement with that determined on the basis of the 1:1 model.

To interpret the above results, some remarks should be made. First, based on the Tz density on the CM3 chip, we estimated an average distance among adjacent antibodies of about 17 nm. As the cap ring of the vault protein has a 13 nm diameter [34], this makes it quite unlikely that an individual vault-Z molecule can interact with more than one Tz at a time, even more when taking into account that conjugation at the chip surface occurs with a random orientation. Thus, only a subpopulation of bound antibodies has a proper orientation for binding. Second, either chain composing the antibody’s Fc portion has a facing-outward binding site for Protein A [35,36]. Of course, this must also hold true when the Protein A-derived Z domain is bound, so it is expected that up to two Z domains can bind to a single antibody molecule. Third, based on Z domain and vault-Z binding affinities (38.3 and 2.53 nM, respectively), it can be determined that the binding free energy values at 25 °C were 42.1 and 49.1 kj mol^−1^, respectively, with the latter figure being well below the one expected when assuming a merely additive contribution of the two binding sites to the overall binding energy. This can be plausibly explained by the fact that the vault cap ring’s geometry is by no means designed for optimal interaction with Fc-binding sites, so the second binding event either occurs occasionally or pays a substantial penalty in terms of entropy loss and/or structural strain. However, the increase in affinity by about an order of magnitude resulting from dual binding plays a substantial role in accomplishing an extremely stable interaction between the two molecular partners, as also documented in the following sections.

### 2.3. Binding Stoichiometry to Vault-Z of Tz and Saturation-Dependent Affinity Decline Are Assessed by Densitometric and Mass Spectrometry Analyses

To explore the actual in-solution binding mode of Tz to vault-Z, we incubated 8 nM of vault-Z (corresponding to 624 nM MVP-Z) with different Tz concentrations, 40 to 320 nM. After a 1 h incubation at 4 °C, the reaction mixes were spun down at 100,000× *g* for 2 h at 4 °C, and suitable amounts of pellets were dissolved in SDS-PAGE sample buffer and subjected to electrophoresis. We found that Tz co-precipitated with vault-Z, but not with authentic vault (Figure 4). Furthermore, Tz and vault-Z were quantified by densitometry analysis by co-electrophoresing the pellets with defined amounts of the two proteins to construct suitable calibration curves (Figure 4). Based on this approach, we could determine the molar ratios of bound antibody to vault-Z (Figure 5).

To confirm the above results, we also performed mass spectrometry. A calibration profile was first created using the relative intensities of Tz and vault-Z peptides resulting from the digestion of mixtures of the two proteins at defined molar ratios (Appendix A). Then, the profile was used to estimate the binding stoichiometries of the conjugates digested under the same conditions.

MS and densitometric data were in satisfactory agreement (Figure 5), showing essentially quantitative antibody binding up to a 10 Tz/vault-Z molar ratio (corresponding to 80 nM Tz). This also matched the expected saturation, as theoretically assessed by inserting a 2.5 nM affinity value into the relevant quadratic equation we developed (Appendix A). Actually, based on this, over 99% of added Tz is predicted to be bound to vault-Z at the binding reaction equilibrium up to 80 nM Tz.

At higher molar ratios, the fraction of bound Tz detected by densitometry and MS declined (Figure 5), although at 160 nM of added antibody, as much as 70% was still bound (corresponding to 14 Tz per vault-Z). At 320 nM Tz, a substantial drop was instead observed, with only 41% of bound antibody. Furthermore, by using the mentioned equation in reverse, we assessed the apparent K_D_ values at the highest molar ratios of Tz/vault-Z, i.e., 20 and 40, thereby obtaining 200 and 694 nM, respectively. At the lowest molar ratios, i.e., 5 and 10, this same assessment was more strongly affected by statistical variability, which made it unfeasible to determine reliable figures under conditions close to quantitative binding.

Overall, the picture emerging from these data suggests that Tz overcrowding at the level of vault-Z cap rings is responsible for the drop in affinity at higher molar ratios. Nevertheless, we could confidently conclude that at least 10 molecules of antibody per vault-Z were irreversibly bound. This is further supported by the complete Tz retention by vault-Z when the nanoparticle (NP) was first bound to the antibody at a molar ratio of 10:1, then pelleted at 100,000× *g*, resuspended after supernatant removal, and again pelleted. When comparing the two pellets, no detectable loss in bound antibody during the process could be appreciated, despite an at least one-hour-long manipulation of the NP and removal of the supernatant along with any possible traces of free antibody (Appendix A).

### 2.4. Trastuzumab and Cetuximab Conjugation with Vault-Z Result in Substantially Enhanced Endocytosis by Target Breast Cancer Cells

We conducted cytofluorimetric analyses to evaluate how vault-Z conjugation with two different monoclonal antibodies (MoAbs) stimulates its endocytosis, also assessing possible dependence of the uptake on the extent of MoAb conjugation. Specifically, Tz and Cetuximab (CTX) dock to Her2 and EGFR receptors, respectively, which are overexpressed in SKBR3 (Her2) and MDA-MB 231 (EGFR) mammary carcinoma cell lines [31,32].

Our experimental setup assumed that each vault-Z NP could tightly bind up to 10–12 antibodies and Fc molecules on the whole, as determined in the present study (Figure 5).

Fluorescent labeling of vault-Z was performed by binding it to Alexa Fluor 488 TFP-conjugated Fc at a molar ratio of vault-Z/Fc of 1:3. Then, the NP was conjugated with Tz at three vault-Z/Tz molar ratios (1:2, 1:5, and 1:10). Furthermore, based on a previously described protocol [37], SKBR3 cells were preincubated for 1 h at 4 °C with either MoAb-free vault-Z (as the control) or conjugated with Tz. Finally, the cells were incubated in basic medium for 20 min at 37 °C to enable the endocytic process.

Data analysis of fluorescence intensities normalized to the cell autofluorescence demonstrated an about 4-fold higher uptake of Tz-conjugated vault-Z by SKBR3 compared with the non-conjugated one, with no significant difference being detected among the three Tz-conjugated vault-Z samples (Figure 6). This latter observation suggests that a 1:2 conjugation molar ratio already allows maximal endocytosis, at least under our working conditions. Of course, this conclusion does not necessarily apply to other cellular targets and targeting antibodies, for which the best conditions must be identified individually.

To also assess targeting specificity, we further assayed MoAb-mediated vault-Z endocytosis by both MDA-MB 231 and SKBR3 cell lines using either CTX or Tz at a 1:5 molar ratio, and otherwise under the conditions adopted in the previous experiment. The results highlighted rigorously selective endocytosis, with vault-Z/Tz and vault-Z/CTX being exclusively taken up by SKBR3 and MDA-MB 231, respectively (Figure 7).

## 3. Discussion

For the past forty years, the vault nanoparticle (NP) has been attracting huge interest for its puzzling and unique biological properties. Plenty of literature has highlighted its pro-survival roles, including stress adaptation and drug resistance, wherein some proteinaceous and ribonucleic vault-associated minor components are also involved in different ways [10].

Regardless of its biological roles, the vault NP, as produced by assembly of the sole MVP, has been exploited as a biotechnological tool suitable for several applications. Mostly (but not exclusively), it has been used as a nano-vector for drug delivery to cancer cells [10,21,38], although other applications have also been recently developed, notably laccase immobilization for decolorization and detoxification of synthetic dye compounds [39].

Besides its huge internal cavity, vault NP’s potential as a nanocarrier is even more boosted by the availability of a protein domain, which is capable of tightly associating with defined MVP binding sites located at the inner side of the vault cavity. This is referred to as the INT domain and occurs naturally at the C-terminus of the vPARP minor vault protein [40]. Thus, by fusing it at the C-terminus of cargo proteins or by otherwise binding it to diverse compounds, they can be taken up and stably incorporated into the nano-assembly. Interestingly, a sheer co-incubation of the latter with INT-tagged proteins results in spontaneous association of the two molecular partners [40,41,42], an occurrence also favored by the dynamic nature of the vault assembly [43,44]. Strikingly, a recent report describes internalization by the vault NP of whole, INT-fused, adeno-associated virus particles, which further highlights its remarkable potential as a carrier of diverse cargo molecules [45].

The scope of vault NP’s biotechnological applications was even more extended by the development of the vault-Z variant that carries the so-called Z domain, 33 residues in length, fused at the C-terminus of the MVP polypeptide chains. Thus, vault-Z NPs expose these domains at both caps, which makes their tight association with monoclonal antibodies (MoAbs) possible [22]. Starting from this experimental background, in the present contribution, we first expressed the vault-Z variant in *K. phaffii*. Then, we characterized the pure NP with respect to (i) MoAb affinity for vault-Z, (ii) possible affinity decline with increasing saturation, and (iii) maximal MoAb binding. Obviously, all of these issues are of paramount importance in view of vault’s antibody-driven, targeted delivery. This is because, on the one hand, MoAbs must be stably associated with a NP to ensure an effective targeted delivery; on the other hand, the ratio MoAb/NP may critically affect the targeting efficiency [46], so it must be precisely controlled.

With this in mind, we used Tz as a model MoAb to assess its affinity for vault-Z using the SPR technology, with Tz being conjugated to the sensor chip. We thus determined values in the 2–3 nM range at a conjugate density on the chip surface only compatible with a 1:1 stoichiometric binding ratio, as also confirmed by binding/release kinetics. Interestingly, under the same working conditions, an about 40 nM affinity was determined for the Z peptide in isolation. As both chains forming the Fc domain in an antibody molecule have a Z peptide binding site, these results make it apparent that the second binding event makes an only marginal contribution to the overall affinity, quite probably due to entropy loss and/or structural strain. Nevertheless, the overall interaction between Fc and Z peptides results in an exceptionally strong, virtually irreversible binding.

We then moved to investigate the actual, in-solution interaction mode between Tz and vault-Z. This was performed by co-incubating Tz at different concentrations with 8 nM of vault-Z (corresponding to 624 nM MVP-Z), ultracentrifuging the resulting mix and determining the amount of the two molecular species found in the pellets by two independent methodologies, i.e., densitometric analyses of SDS-PAGE gels and mass spectrometric quantification, which provided reasonably consistent results. They showed, in particular, essentially quantitative Tz binding up to a molar ratio of Tz/vault-Z of at least 10, whereas at higher molar ratios, the fraction of bound Tz declined. On the whole, this experimental pattern points to an extremely high affinity at lower molar ratios (i.e., 5 and 10), in keeping with the K_D_ value provided by SPR measurements, whereas at higher ratios, a substantial drop in apparent affinity and fraction of bound Tz was detected. This is quite likely to be ascribed to antibody overcrowding at the level of the vault-Z caps. Thus, these constraints should be taken into account when conjugating vault-Z with antibodies in view of its targeted delivery.

Starting from the above findings, we then extended our investigation by using the IgG1 Fc portion [22,24] as a scaffold to be conjugated with different molecules before association with vault-Z. Actually, the sole Fc portion also binds to the Z peptide as tightly as antibodies, so in the present work, we exploited it as a carrier of a fluorophore (Alexa Fluor 488 TPF) to label vault-Z in cytofluorimetric analyses designed to monitor MoAb-driven vault-Z uptake by the mammary carcinoma cell lines SKBR3 and MDA-MB 231.

In such investigations, we observed that vault-Z endocytosis was substantially stimulated when the NP was conjugated with both MoAbs and, concurrently, the process was proven to be rigorously selective. Actually, in line with their well-known specificity, Tz and CTX targeted SKBR3 and MDA-MB 231 cells, respectively, which in turn overexpressed either Her2 (SKBR3) or EGFR (MDA-MB) [47,48].

From a wider perspective, the Fc-based experimental layout described here substantially broadens the application scope of the vault-Z NP, as it allows both conjugation with a repertoire of diverse molecules, besides antibodies, and fine adjustment of the number of bound ligands, just as in the case of antibodies. No less important, it also streamlines the labeling procedures by minimizing vault-Z manipulation, which inevitably results in decreased recovery. Among candidate molecules to be associated with the NP via Fc, it is worth mentioning, besides fluorophores, cell-targeting peptides [30], and/or the pVI peptide, which enables endosomal escape [49], and/or anticancer drugs, as we are planning to do in our future investigations.

## 4. Materials and Methods

### 4.1. Strain and Growth Media

For the production of the MVP protein, the vacuolar aspartyl protease PEP4-deficient *K. phaffii* SMD1168 (his4, ura3, pep4::URA3) strain was used [50]. The strain is available from Invitrogen (ThermoFischer Scientific, Waltham, MA, USA). *K. phaffii* was grown in a flask in BMDY medium (10 g/L yeast extract, 20 g/L bacto peptone, 20 g/L dextrose, 100 mM potassium phosphate buffer, pH 5.8, 13.4 g/L yeast nitrogen base without amino acids, and 0.4 mg/L biotin). All media were from Biolife, while monobasic potassium phosphate, yeast nitrogen base without amino acids, and biotin were from Sigma-Aldrich (Saint Louis, MO, USA). Standard liquid and plate growth were performed on YPD medium (10 g/L yeast extract, 20 g/L bacto peptone, 20 g/L dextrose, and 20 g/L agar omitted in liquid medium) with or without 100 μg/mL of zeocin. After electroporation, YPDS plates containing 1 M sorbitol and 100 μg/mL of zeocin were used for the selection of the transformants.

### 4.2. MVP-Z Gene Cloning and Selection of Positive Clones

Authentic human MVP gene cloning and selection of the best-producing clones has been previously described [25]. The MVP gene variant encoding the Z peptide at the C-terminus (MVP-Z) was cloned, as follows. The C-terminal sequence of MVP was removed from the previously constructed [25] pGAPZ B MVP vector by digestion with *Sac*I and *Not*I. Electrophoresis of the digestion mix on 1% agarose gel using TAE as a running buffer was run for 45 min at 90 V and gel-stained with EtBr, which confirmed the digestion of the plasmid. The *Not*I/*Sac*I-digested pGAPZB carrying the N-terminal sequence of MVP was purified from gel using the QIAquick^®^ Gel Extraction Kit (QIAGEN, Germantown, MD, USA) and quantified with NanoDrop (NanoDrop 2000c Spectrophotomer, ThermoFischer Scientific). The Z peptide amino acid sequence to be added to MVP C-terminus was: FNMQQQRRFYEALHDPNLNEEQRNAKIKSIRDD [24]. The Z peptide-encoding nucleotide sequence (purchased from Eurofins and reported in Appendix A) carried sticky ends complementary to those resulting from *Not*I/*Sac*I digestion of the pGAPZB plasmid and exposing the MVP C-terminal sequence, which allowed in-frame fusion and plasmid re-circularization. Cloning was achieved using the NEBuilder^®^ HiFi DNA Assembly Master Mix (New England Biolabs, County Road, Ipswich, MA, USA). The sequence was confirmed by Sanger sequencing, commissioned to Eurofins. Transformation of *K. phaffii* and selection of MVP-Z-positive clones were performed as previously reported when handling authentic MVP [25].

### 4.3. Cell Growth, Collection, and Cell-Free Extract Production

Growth of the yeast clone expressing authentic vault and subsequent production of cell-free extracts was performed as previously described [25]. Likewise, production of vault-Z-containing extracts was achieved following the same procedure.

### 4.4. Vault Purification

The purifications of both authentic vault and vault-Z were carried out according to the procedure we previously developed, essentially consisting of RNase treatment of cell-free extracts, subsequent centrifugation to remove ribosomal contaminations, followed by size exclusion chromatography (SEC) using Sepharose CL-6B as the matrix [25]. Eluted protein was assayed by the bicinchoninic acid assay using the kit QPRO-BCA (Cyanagen, Bologna, Italy) and bovine serum albumin as the calibration standard.

### 4.5. Electrophoretic Analyses

The purified vault-Z samples and vault-Z-antibody complexes were analyzed by SDS-PAGE. Typically, 6–12 μg samples were applied. Gels were stained with Imperial Protein Stain (ThermoFisher Scientific), and PageRuler Unstained Protein Ladder (ThermoFisher Scientific) was used for the molecular weight calibration. MVP identity was confirmed by Western blot analysis on the PVDF membrane Immobilon (Merck Millipore, Burlington, MA, USA) using a primary anti-MVP antibody (rabbit monoclonal, 1:10,000 dilution in PBS, 5% skim milk; Abcam, Cambridge, UK) and a fluorescent secondary anti-rabbit IR-800 antibody (1:16,000 dilution in PBS, 5% skim milk; LI-COR Biosciences, Lincoln, NE, USA). The MVP signal was revealed using an Odyssey FC instrument (LI-COR Biosciences).

### 4.6. Transmission Electron Microscopy Analysis

Transmission electron microscopy (TEM) images of vault particles were obtained using a Tecnai12 (RH42B) microscope (accelerating voltage: 120 kV—filament: LaB6) on the *ImagoSeine* platform, Institute Jacques Monod (Paris, France). Sample preparation and analysis were conducted as previously reported [25]. Samples were deposited on carbon-coated copper grids, 400 mesh, after plasma activation for 20 s, by floating the grid onto the protein drop (20 μL, 0.5 mg/mL) for 1–2 min. The grid was then dried from liquid excess by filter paper and put on a drop of uranyl acetate (1% in PBS, pH 5.0) for 1–2 min, depending on the sample concentration. Finally, the grids were dried with Whatman filter paper and analyzed.

### 4.7. Dynamic Light Scattering Analysis

Dynamic light scattering (DLS) measurements were performed by a Zeta Sizer Nano Instrument (Malvern Instruments Ltd., Malvern, Worcestershire, UK) operating at 4 mW of a HeeNe 633 nm laser, using a scattering angle of 90°. A disposable cuvette with a 1 cm optical path length was used for the measurements. Before analysis, the samples were spun down at 20,000× *g* for 20 min, so they were cleared of any turbidity. Then, they were allowed to equilibrate for 2 min prior to measurement. Routinely, sample concentration was in the range of 0.2–0.5 mg/mL. Three independent measurements of 60 s duration were performed at 25 °C. Calculations of the hydrodynamic diameter were performed using the Mie theory, considering the absolute viscosity and the refractive index of the material set to 1.450, Abs 0.001. The number-based hydrodynamic diameter and the autocorrelation function were determined, with the latter being diagnostic of sample homogeneity, insofar as a single exponential decay profile was detected.

### 4.8. Surface Plasmon Resonance for the Determination of Z Peptide-Antibody and Vault-Z-Antibody Binding Affinities

A Biacore X100 system (Cytiva-Pall, Marlborough, MA, USA) was used to perform SPR-based analyses of molecular interactions. The Tz antibody was coupled to a carboxymethylated dextran surface of a CM3 sensor chip by using amine-coupling chemistry. The amine-coupling procedure was performed setting the instrumentation temperature at 25 °C and using the running buffer HBS-EP+ (10 mM HEPES, pH 7.4, 0.15 M NaCl, 3 mM EDTA, and 0.05% *v*/*v* Surfactant P20) at a flow rate of 5 μL/min and in the following three steps, as recommended by the producer (Biacore Sensor Surface Handbook BR100571) and adapted for our purposes. First, the CM3 chip was activated by injecting EDC/NHS (1/1) on both flow cells 1 and 2 for 7 min; second, Tz was diluted in 10 mM sodium acetate, pH 4.5, at a final concentration of 20 μg/mL and injected on the flow cell 2 until reaching a surface density of 1220 RU; third, 1 M ethanolamine-HCl, pH 8.5, was injected on both cells. Appropriate, multiple concentrations of authentic vault, vault-Z, and peptide Z (ProteoGenix, Schiltigheim, France) were injected for 3 min at 25 °C and at a flow rate of 30 μL/min in running buffer (10 mM HEPES, pH 7.4, 0.15 M NaCl, 3 mM EDTA, and 0.05% *v*/*v* Surfactant P20). To preserve the protein stability, vault-Z and authentic vault were stored in ice before injection, then injected at room temperature. After injection, analyte solutions were replaced by running buffer at a continuous flow rate of 30 μL/min. Surface regeneration was accomplished by injecting 50 mM NaOH for a contact time of 1 min. Each sensorgram was subtracted for the response observed in the control flow cell (no immobilized protein) and normalized to a baseline of 0 RU. The sensorgram profiles were acquired by setting the Biacore X100 Control software, version 2.0.2 (Cytiva-Pall, Marlborough, MA, USA), in manual run mode for authentic vault, vault-Z, and immobilization procedure injections, and in multi-cycle kinetics mode for peptide Z. The interaction rate constants were calculated by fitting the sensorgrams to the Langmuir Binding 1:1 model and using the BIA Evaluation software 2.0.2 (Cytiva-Pall, Marlborough, MA,) for Peptide Z and BIA Evaluation 4.1 software (GE Healthcare Life Sciences, Chicago, IL, USA) for vault-Z.

### 4.9. Densitometric and Mass Spectrometric Determination of Vault-Z-Antibody Binding Stoichiometry

Different amounts of antibody were mixed with a fixed amount of vault-Z and the molar ratios of bound antibody to vault-Z were then determined, as follows. Briefly, 8 nM of vault-Z (corresponding to 624 nM of MVP-Z) was incubated with scalar concentrations (40 to 320 nM) of Trastuzumab (Tz) at 25 °C for 1 h in 12 mM sodium phosphate, pH 7.2, 0.14 M NaCl, and 2.7 mM KCl, in a final volume of 7.5 mL. Then, the mixes were centrifuged for 2 h at 4 °C and 100,000× *g*. The pellets were dissolved in SDS-PAGE sample buffer and suitable amounts were subjected to electrophoresis. The gels were stained with Imperial Protein Stain (ThermoFisher Scientific) and images were acquired by an Odyssey FC instrument (LI-COR Biosciences, Lincoln, NE, USA). Vault-Z and Tz protein contents of the individual bands were determined densitometrically using the Image J software (version 2.9.0/1.53t) and scalar amounts of purified vault-Z and Tz (quantified by the BCA assay) to construct the calibration curves. The same procedure was used for the assessment of Tz retention upon two ultracentrifugation procedures (Appendix A).

The binding stoichiometry of Tz conjugated to MVP-Z was also assessed by reverse-phase liquid chromatography/mass spectrometry (LC/MS). For this purpose, a calibration curve was first constructed by mixing vault-Z and Tz at predefined molar ratios, reducing, alkylating, and digesting the mixtures by a standard in-solution protocol for MS [51]. The resulting peptides were then desalted by a C18 zip-tip column (Merck-Millipore, Burlington, MA, USA) and injected in an Orbitrap Fusion mass spectrometer coupled with an EASY-1000 LC system (ThermoFischer Scientific). The peptides were separated by a 50 cm C18 EASY-Spray column (ThermoFisher Scientific) employing a 1 h gradient (0–80% acetonitrile), detected in the orbitrap analyzer and fragmented by high-energy collision-induced dissociation (HCD) in the ion-trap analyzer of the instrument. The peptides were identified and quantified by a label-free approach using the Proteome Discoverer software 2.3 (ThermoFisher Scientific). Based on signal intensities and missed cleavages rates, six peptides for each protein were selected to create the calibration curve for the MVP-Tz relative quantification. To assess binding stoichiometries of co-incubated Tz and vault-Z, the incubation mixtures were previously collected by ultracentrifugation at 100,000× *g* for 2 h at 4 °C, dissolved in SDS sample buffer (0.4% SDS), then subjected to the above procedure, using the calibration curve to quantify the two molecular partners.

### 4.10. Vault-Z Conjugation with MoAbs and Fluorescently Labeled Fc

Alexa Fluor 488 TPF (ThermoFisher Scientific) was added to 0.9 mg of Fc human antibody portion (Athens Research and Technology, Athens, GA, USA) in 20 mM of potassium phosphate, 0.137 M NaCl, pH 8.0, at a molar ratio of 10:1 (10 mg/mL of fluorophore stock solution in DMSO) in a 1 mL volume and incubated for 1 h at room temperature under shaking. The reaction mixture was purified from unreacted dye in Zeba™ Dye and Biotin Removal Spin Columns (ThermoFisher Scientific). Fluorescent-conjugated Fc (Fc488) was added to 1 mg of vault-Z in phosphate buffer saline (PBS: 0.14 M NaCl, 27 mM KCl, 100 mM Na_2_HPO_4_, and 10 mM KH_2_PO_4_) at a molar ratio of 3:1 in 2.7 mL and incubated for 1 h at 4 °C under shaking. Then, 0.25 mg of the resulting vault-Z-Fc488 was conjugated with either Tz at different vault-Z-Fc488/Tz molar ratios (1:2, 1:5, and 1:10) or CTX at the sole 1:5 molar ratio. The reaction was carried out in volumes in the range of 0.6–1 mL for 1 h at 4 °C under shaking. The resulting conjugate concentration was then determined by UV VIS, using a 2000/2000c NanoDrop Spectrophotometer (ThermoFisher Scientific). Both MoAb-free and conjugated vault-Z-Fc488 were stored overnight at 4 °C, then diluted in basic buffer (DMEM with 0.1% BSA, 20 mM Hepes, pH 7.4) to a final concentration of 20–25 μg/mL (binding buffer) immediately before their use in endocytosis experiments.

### 4.11. Analysis of Vault-Z Endocytic Uptake after MoAb Conjugation

To quantify both MoAbs-free and bound vault-Z-Fc488 internalization in cancer cells, cytofluorimetric analyses were conducted on the mammary carcinoma cell lines SKBR3 and the triple-negative MDA-MB 231 Luc+, purchased from ATCC. SKBR3 cells were cultured in 50% Dulbecco’s Modified Eagle’s Medium High Glucose (DMEM-HG) and 50% HAM’S F12. MDA-MB 231 Luc + cells were cultured in 100% DMEM-HG with 1% of non-essential amino acids solution (MEM NEAA, 100×). Both media were supplemented with 10% fetal bovine serum, 2 mM L-glutamine, 50 IU/mL penicillin, and 50 mg/mL streptomycin, and cells were incubated at 37 °C in a humidified atmosphere containing 5% CO_2_ and sub-cultured using trypsin/EDTA prior to confluence.

Cytofluorimetric analyses were performed following a previously described protocol [37], with minor modifications. Briefly, 3 × 10^5^ cell aliquots were seeded in a 12-well plate and incubated for 40 h (SKBR3) or 24 h (MDA-MB 231) at 37 °C. Cells were then preincubated for 4 h with 1 mL/well of basic buffer at 37 °C. Subsequently, the basic buffer was removed, and 1 mL/well of ice-cold binding buffer (DMEM with 0.1% BSA, 20 mM Hepes, pH 7.4) containing 20–25 μg/mL of vault-Z-Fc488 or MoAb-bound vault-Z-Fc488 was added. Cells were incubated with binding buffer for 1 h at 4 °C under gentle shaking. Cold medium was replaced by 1 mL/well of warm basic buffer and cells were transferred to 37 °C for 20 min. Afterwards, they were washed twice with ice-cold PBS with Ca^2+^ and Mg^2+^ and incubated with acidic wash buffer (0.2 M acetic acid, pH 2.8, 0.5 M NaCl) for 5 min at 4 °C under gentle shaking, to remove surface-bound protein. After incubation, cells were washed with PBS without Ca^2+^ and Mg^2+^, then with trypsin-EDTA to detach them. Complete medium was added, and samples were centrifuged at 1300 rpm for 5 min. Pellets were washed twice with PBS without Ca^2+^ and Mg^2+^ and resuspended in 0.5 mL of PBS with 2 mM of EDTA. Flow cytometer analysis was performed with the CytoFLEX Flow Cytometer (Beckman coulter, Brea, CA, USA) and samples were analyzed using the CytExpert Software 2.4 (Beckman coulter).

## 5. Conclusions

The biotechnological platform based on the vault-Z NP developed in the present investigation offers remarkable advantages as a drug delivery tool. We determined that it can irreversibly bind up to at least ten antibodies and/or Fc antibody domains on the whole. Thus, one can finely adjust the molar ratio antibody/NP in view of optimized targeting. No less important, the Fc domain can be exploited as a scaffold to be conjugated with different molecules, such as fluorophores, as we performed in the present investigation, but also cell-targeting peptides. As in the case of antibodies, the molar ratio of Fc/NP can also be finely adjusted along with the associated molecules, be it fluorophores or peptides. Taken together, these technological developments substantially broaden the scope of this nanodevice, as compared with more conventional approaches when designing protocols of targeted delivery.

## Figures and Tables

**Figure 1 ijms-25-06629-f001:**
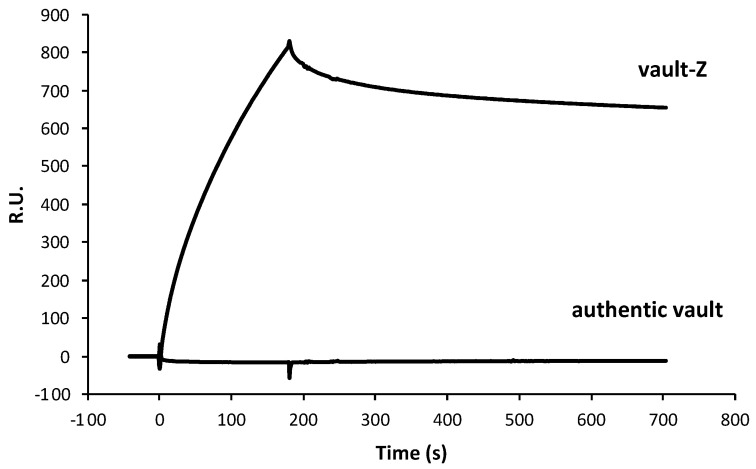
Association–dissociation binding kinetics between either the authentic vault or vault-Z (50 nM) and Tz. The two proteins were injected in running buffer over the CM3 chip with immobilized Tz antibody, then displaced by injecting the sole running buffer.

**Figure 2 ijms-25-06629-f002:**
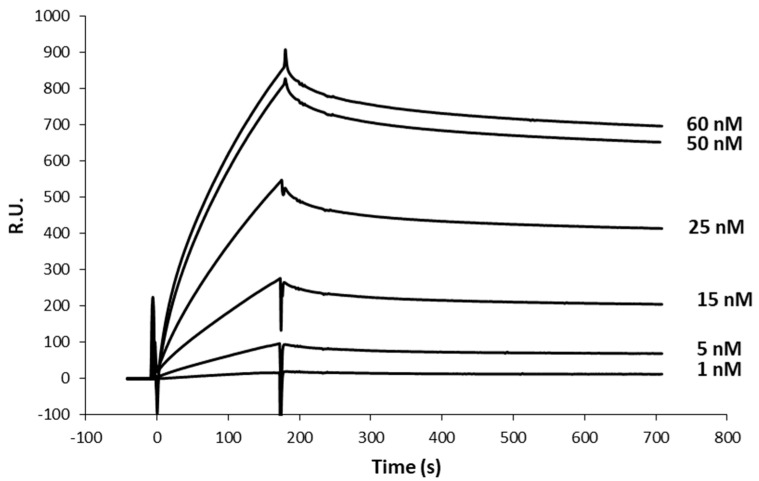
Association–dissociation binding kinetics between vault-Z and Tz. Vault was injected in running buffer at growing concentrations, as indicated in the figure, then displaced by injecting the sole running buffer.

**Figure 3 ijms-25-06629-f003:**
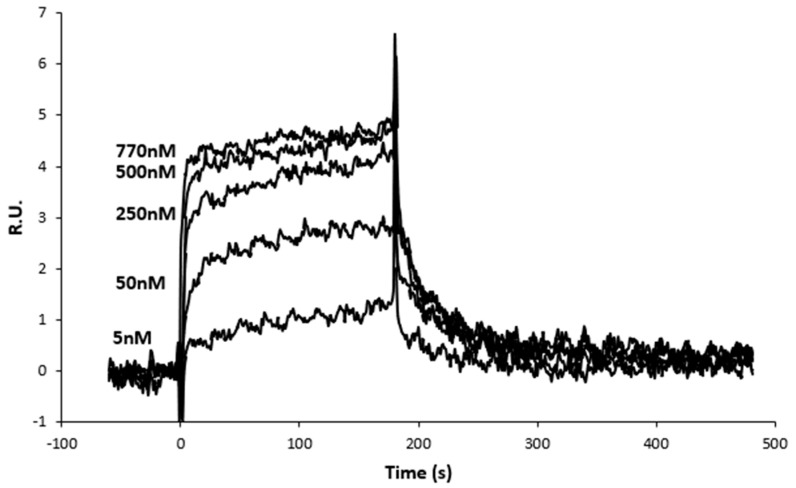
Association–dissociation binding kinetics between Z peptide and Tz. Z peptide was injected in running buffer at growing concentrations, as indicated in the figure, then displaced by injecting the sole running buffer.

**Figure 4 ijms-25-06629-f004:**
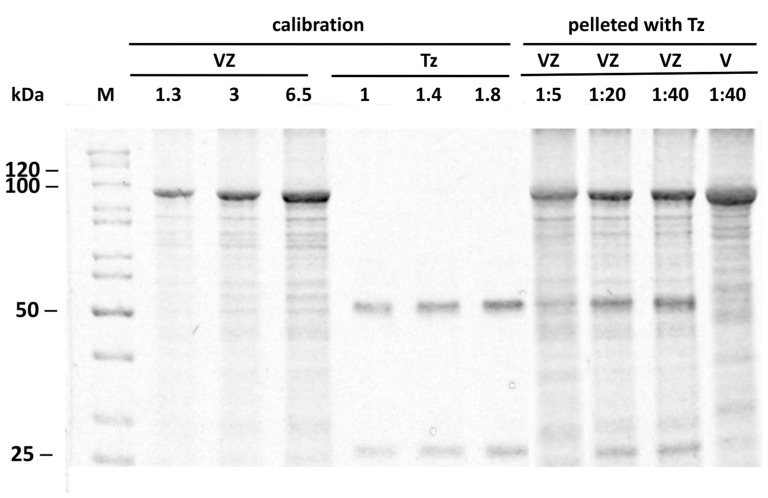
A representative co-electrophoresis of pure vault-Z (VZ) and Trastuzumab (Tz) samples along with VZ or authentic vault (V) co-incubated with Tz and pelleted at 100,000× *g*. Defined amounts of VZ and Tz (expressed in μg) were subjected to SDS-PAGE (12% gel) to construct calibration intensity profiles. In the same gel, pelleted incubation mixtures of VZ/Tz at the indicated molar ratios were run. As a control, authentic vault (V) was also run. M: unstained PageRuler protein ladder (ThermoFisher Scientific, Waltham, MA, USA) with the respective molecular weights (kDa). Gels were stained using Imperial Protein Stain (ThermoFisher Scientific). Other details are provided in the Material and Methods (Section 4.9).

**Figure 5 ijms-25-06629-f005:**
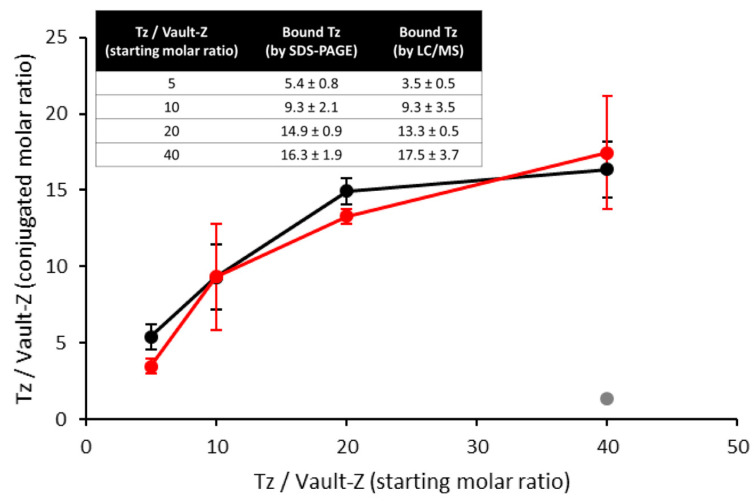
Binding stoichiometry of Tz/vault-Z conjugates as a function of in-solution molar ratios, as assessed by either SDS-PAGE densitometry (black profile) or LC/MS (red profile). Error bars correspond to the standard deviation over three (SDS-PAGE) or four (LC/MS) independent experiments. The gray circle represents the negative control performed by LC/MS analysis of authentic vault (i.e., devoid of the Z peptide). Inset: in-solution versus bound Tz at the adopted Tz/vault-Z molar ratios, as determined by both densitometry and LC/MS.

**Figure 6 ijms-25-06629-f006:**
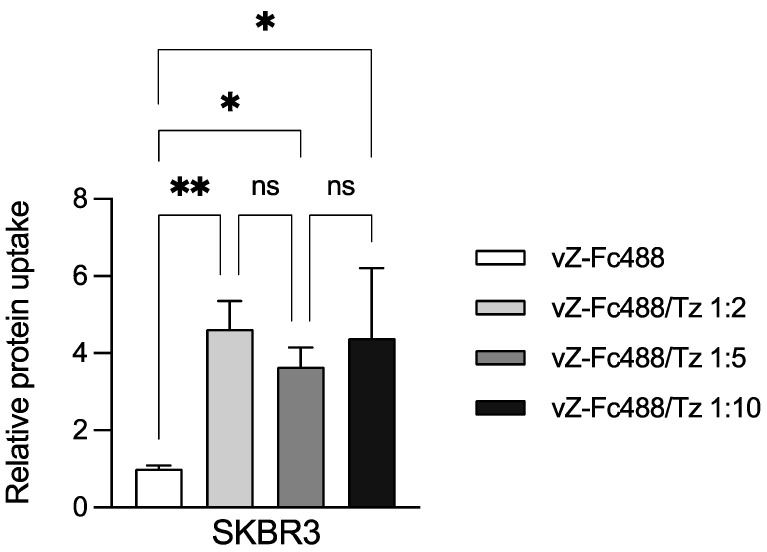
Uptake by the human breast cancer SKBR3 cell line of Tz-conjugated vault-Z. The NP was first labeled with Fc-Alexa Fluor 488 TFP, then bound to Tz at the indicated molar ratios. Cells were subsequently incubated with Tz-free or Tz-bound vault-Z in complete medium. Measurements were normalized to the autofluorescence emission in the absence of vault-Z and otherwise under the same conditions. Data are the mean ± standard deviation of three biological replicates, each of them being obtained from the mean of three technical replicates. Statistical analyses were performed using the one-way ANOVA test. * *p* < 0.05; ** *p* < 0.01; ns: statistically not significant. Other details are provided in Section 4.10 and Section 4.11.

**Figure 7 ijms-25-06629-f007:**
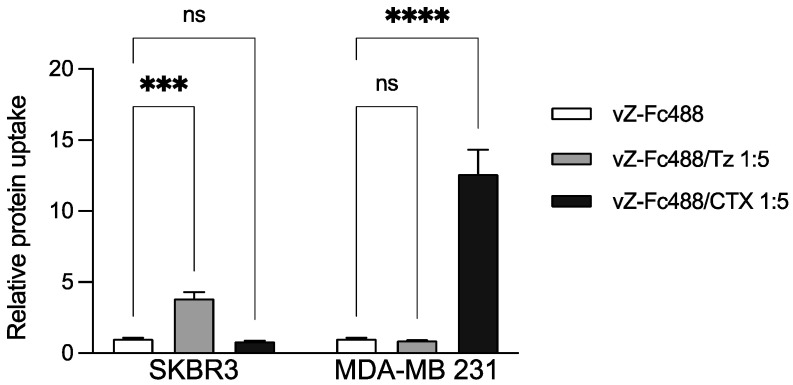
Uptake of Tz- and CTX-conjugated vault-Z by the human breast cancer cell lines SKBR3 and MDA-MB 231. The NP was first labeled with Fc-Alexa Fluor 488 TFP, then bound to both MoAbs at a 1:5 molar ratio. Cells were subsequently incubated with MoAb-free or MoAb-bound vault-Z. Measurements were normalized to the autofluorescence emission in the absence of vault-Z and otherwise under the same conditions. Data are the mean ± standard deviation of three biological replicates, each of them being obtained from the mean of three technical replicates. Statistical analyses were performed using the two-way ANOVA test. *** *p* < 0.001; **** *p* < 0.0001; ns: statistically not significant. Other details are provided in Section 4.10 and Section 4.11.

**Table 1 ijms-25-06629-t001:** Rate and equilibrium constants ± standard deviations of Tz binding to either vault-Z or the Z-domain in isolation, as determined by SPR analysis.

Protein	k_a_ (10^4^ 1/Ms)	k_d_ (10^−4^ 1/s)	K_D_ (nM)
Vault-Z	10.5 ± 5.0	2.59 ± 0.2	2.53 ± 0.4
Z-domain	44.6 ± 13.6	171 ± 84	38.3 ± 23.5

## Data Availability

Not applicable.

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
