# Peer review of "An Efficient Method for Vault Nanoparticle Conjugation with Finely Adjustable Amounts of Antibodies and Small Molecules"

_ijms, 2024, doi:10.3390/ijms25126629_

Round 1

Reviewer 1 Report

Comments and Suggestions for Authors

This manuscript, “An efficient method for vault nanoparticle conjugation with finely adjustable amounts of antibodies and small molecules”, describes the production and characterization of a recombinant empty human vault particle with an Fc-binding peptide (the “Z“ domain) expressed at the C-terminus of the major vault protein (MVP). Thus, the assembled vault has 39 Z domains displayed at the top and bottom caps of the particle. Although the production and targeting of the vault-Z was previously described, this manuscript does an excellent job of characterizing the Fc binding kinetics and the utilization of a fluorescent Fc fragment to follow specific targeting of the antibody-conjugated vault to two different cancer cells in vitro.

Although an incremental advance, I believe this manuscript makes a significant contribution to our understanding of the use of vault-Z for nanoparticle targeting. The binding kinetic data are solid and clearly presented, as are the targeting experiments, although it would have been even more interesting to analyze vault-Z antibody conjugates in an in vivo animal tumor model.

I have only one significant criticism and a few minor suggestions.

First my major concern. In the supplemental information, Figure S1 panel C, the authors show four example images of the purified vault-Z nanoparticles stained with uranyl acetate. These images trigger my concerns, as the staining that is seen is somewhat unexpected. Two of the vaults have large (30 to 50 nm) densities emanating from one end of the particle and two of the images do not show any extra density at all at one end of the particle. As 39 Z-domains are present on each vault cap these represent approximately 14,000 Da of extra mass. If these are truly representative images, then I am puzzled how the Z-domain is behaving in three-dimensional space. Are these 30 to 50 nm densities truly representative of the added mass? Perhaps a related piece of data is in S1, panel B. Here a DLS analysis is presented that states in the caption that 99.4% is a major component that is 52.27 +/- 6.21 nm and the minor component (I assume 0.6%) that is 191.00 +/- 43.08 nm. Does this mean that some particles are huge (191 nm) because they have this extra density at both ends? I think the authors should directly address this point. They should show a low magnification field of the stained vault-Z particles (this can be in the supplemental section) so that dozens of examples can be seen and examined. I would also like the authors to address the apparent particle heterogeneity seen in the negative staining, I think this could be important. Is it due to the flexible MVP C-termini?

Minor concerns:

1.     The authors state that vault function is “mainly cytoprotective”. My understanding of the vault literature is that, although there have been some publications that suggest this function, no follow up or supportive data have been presented to confirm these hypotheses, and therefore any conclusions about vault function remain conjecture.

2.     Figure 1.  “Authentic vault” in my opinion would refer to a native vault made in a eukaryotic cell. It may be more accurate to refer to the vault in this figure as a “recombinant vault shell” or a “recombinant empty vault”.

3.    The title of the manuscript is a bit misleading as it over promises. What small molecules besides Fc were demonstrated?

Comments on the Quality of English Language

 English is likely not the native language of the authors of this manuscript, and I greatly appreciate their efforts to write clearly. However, the manuscript would be significantly improved by an editor correcting the awkward word usage, and sentence structure throughout.

Author Response

MAJOR CRITICISM:

First my major concern. In the supplemental information, Figure S1 panel C, the authors show four example images of the purified vault-Z nanoparticles stained with uranyl acetate. These images trigger my concerns, as the staining that is seen is somewhat unexpected. Two of the vaults have large (30 to 50 nm) densities emanating from one end of the particle and two of the images do not show any extra density at all at one end of the particle. As 39 Z-domains are present on each vault cap these represent approximately 14,000 Da of extra mass. If these are truly representative images, then I am puzzled how the Z-domain is behaving in three-dimensional space. Are these 30 to 50 nm densities truly representative of the added mass? Perhaps a related piece of data is in S1, panel B. Here a DLS analysis is presented that states in the caption that 99.4% is a major component that is 52.27 +/- 6.21 nm and the minor component (I assume 0.6%) that is 191.00 +/- 43.08 nm. Does this mean that some particles are huge (191 nm) because they have this extra density at both ends? I think the authors should directly address this point. They should show a low magnification field of the stained vault-Z particles (this can be in the supplemental section) so that dozens of examples can be seen and examined. I would also like the authors to address the apparent particle heterogeneity seen in the negative staining, I think this could be important. Is it due to the flexible MVP C-termini?

REPLY

Regarding the criticism on TEM images, we have browsed many of those we acquired after uranyl acetate staining of vault-Z. In the revised version, we have replaced the previous images with new ones that in our opinion better help address the Reviewer’s concern. Furthermore, we have also acquired further images using lanthanum-based uranyl acetate replacement stain (attached). On the whole, an accurate analysis of these observations makes it possible to draw the following conclusions:

1) the large densities emanating from vault’s ends are non-protein artifactual clumps. This is substantiated by the following considerations: a) they also occur in isolation, as detected in the new Fig. S1; b) only occasionally are they observed in association with vault-Z, never being detected at both sides of a given nanoparticle. This is also attested by lanthanum-stained samples, where such clumps do not appear. We prefer however not to include in the paper these images, due to poor contrast; c) they are distinctly different from vault-Z in their appearance, as they look lighter in color; d) regarding the size of the C-terminal Z-peptides, taken together their molecular mass amounts to about 140 kDa (i.e. 3600 x 39), which is about 2% of the total vault-Z mass. Such a size cannot be seen in a TEM image, as also in the case of IgGs (Mr: 150 kDa).

2) concerning the issue related to the DLS spectra, in all likelihood the 0.6% component is representative of vault-Z clusters, given the well-known propensity of the nanoparticle to aggregate when the working conditions are not accurately controlled (see ref. 25). On the other hand, an entity with an average 191 nm can only arise from the association of several vault nanoparticles.

MINOR CONCERNS:

  1. The authors state that vault function is “mainly cytoprotective”. My understanding of the vault literature is that, although there have been some publications that suggest this function, no follow up or supportive data have been presented to confirm these hypotheses, and therefore any conclusions about vault function remain conjecture.

We admit that, based on the available literature, a cytoprotective action can be assigned to the vault nanoparticle just on a speculative basis. However, it also true that vault’s role in multi-drug resistance seems to be well established. And MDR is a kind of cytoprotection, albeit “sui generis”. In any case, we have deleted this word throughout, yet leaving “prosurvival”, which may be more representative of its actual biological roles.

  1. Figure 1. “Authentic vault” in my opinion would refer to a native vault made in a eukaryotic cell. It may be more accurate to refer to the vault in this figure as a “recombinant vault shell” or a “recombinant empty vault”.

The expression “authentic vault” wants to mean: “a recombinant molecule, yet identical to its natural counterpart”. In the context of our work, defining vault-Z as “recombinant” might be misleading as both vault and vault-Z are recombinant. In any case, to comply with the Reviewer’s recommendation and make clearer the concept, we have specified at the in the Sections Introduction and Results (lines 80 and 112, respectively): “…authentic vault (i.e., devoid of the Z peptide)”.

  1. The title of the manuscript is a bit misleading as it over promises. What small molecules besides Fc were demonstrated?

Unquestionably, as far as the present work is concerned, the procedure we developed was used to just conjugate the vault with a fluorophore via the Fc domain. However, this strategy discloses much wider and promising perspectives, as pointed out in the discussion, just because many other molecules can be conjugated to the vault following the same straightforward procedure. In our opinion this is one of the most significant contribution provided by the present work, albeit in prospective, so we think this should be in some way highlighted in the title.

- English is likely not the native language of the authors of this manuscript, and I greatly appreciate their efforts to write clearly. However, the manuscript would be significantly improved by an editor correcting the awkward word usage, and sentence structure throughout.

English has been thoroughly revised by a professional.

Reviewer 2 Report

Comments and Suggestions for Authors

Comment

Date: 05-05-2024

Manuscript ID: ijms-3014206

Tomainoet al. addressed very interesting findings in their research entitle as “An efficient method for vault nanoparticle conjugation with finely adjustable amounts of antibodies and small molecules”. The work is interesting and informative to reader working in the domains. However, I recommend few suggestions before publication.

Comment 1: In abstract, I suggest to describe exact particle size (nm). Three dimensions (72.5 x 41 x 41 nm in size) is related to volume. This may create confusion to readers. Moreover, nanoparticles and nanocapsules are quite different drug delivery systems. Therefore, I suggest to use anyone of them. Please define “breast cancer cell lines.” In abstract.

Comment 2: In introduction section, the sentence “that are capable of binding molecular receptors overexpressed at the cell surface”. I recommend to include information regarding the overexpressed receptors on the tumor cells. Please name and their biological functionality to interact with the conjugate or Mab.  

Comment 3: In section 4.1, I did not find the source of K. phaffii SMD1168. What was the incubation time, temperature, and CFU used for the study?. How did the authors avoid the cross contamination during culture growth and incubation?. There may be chances of cell death due to lack of sufficient humidity required for the culture growth. How did you maintain? In the sentence “the grids were dried with Whatman filter paper and analyzed.”, it is unavoidable handling of the sample for TEM analysis. I suggest to remove it.

Comment 4: In section 4.6, TEM analysis was conducted to observe morphological assessment. The culture visualization after drying is a challenging task for real image scanning. How did the authors manage to avoid heat induced culture damage and structural changes?. On the grid, there were several microscopic pores to trap the loaded sample. How the preferential adsorption of small size culture did the author prevent?

Comment 4: In section 4.7, the authors analyzed size. Please mention the nature of sample (transparent or turbid). Did the author dilute the sample before analysis? If yes, please include the details.  

Comment 5: For the sentence “the mammary carcinoma cell lines SKBR3 535and the triple negative MDA-MB 231 Luc+, purchased from ATCC. SKBR3 cells were cul- 536tured in 50% Dulbecco’s Modified Eagle’s Medium High Glucose (DMEM-HG) and 50% 537HAM’S F12. MDA-MB 231 Luc + cells were cultured in 100% DMEM-HG with 1% of non- 538essential amino acids solution (MEM NEAA, 100X).” I recommend to shift this in material section. I found repetition of various materials with source in the text body.

Comment 6: I suggest to replace figure 3 with fine line of curve setting. Figure 4 is not clear. I recommend to provide high resolution image of figure 4.   

Comment 7: trade mark is copyright related to owner of the material or tool. I suggest to remove the superscript of “TM” as present in “PageRuler™”.

Comment 8: In figure 5, standard error values are exceptionally higher than the mean. Please check for the value. It must not be too high.  

Comment 9: I found the plagiarism report at 51%. It cannot be accepted for publication in this form.

Comments on the Quality of English Language

Minor

Author Response

REVIEWER 2

COMMENT 1: In abstract, I suggest to describe exact particle size (nm). Three dimensions (72.5 x 41 x 41 nm in size) is related to volume. This may create confusion to readers. Moreover, nanoparticles and nanocapsules are quite different drug delivery systems. Therefore, I suggest to use anyone of them. Please define “breast cancer cell lines.” In abstract.

1. To comply with the Reviewer’s recommendation we have replaced the former sentence (a nanocapsule, 72.5 x 41 x 41 nm in size) with the following: “a nanoparticle with an about 60 nm volume-based size” (line 12). Regarding the term “nanocapsule”, we acknowledge it does not properly apply to the case of the vault NP. So, we have replaced it throughout by the more general term nanoparticle (abbreviated: NP). Also, we have explicitly designated the cancer cell lines, as requested (line 23).

COMMENT 2: In introduction section, the sentence “that are capable of binding molecular receptors overexpressed at the cell surface”. I recommend to include information regarding the overexpressed receptors on the tumor cells. Please name and their biological functionality to interact with the conjugate or Mab. 

Regarding this recommendation, we would like to draw the Reviewer’s attention on the fact that this sentence illustrates a general feature of MoAbs employed in clinical and cellular cancer research. So, in this context it is not appropriate to mention specific receptors. In contrast, we already mentioned explicitly those relevant to our research in the sections Results and Discussion, along with the respective overexpressing cell lines (lines 249-251 and 360-362, respectively).

COMMENT 3: In section 4.1, I did not find the source of K. phaffii SMD1168. 

This strain is produced by Invitrogen, although it is commercialized by ThermoFisher. Invitrogen was already mentioned in the manuscript (in the revised version see line 376). In the revised version we also mention ThermoFisher (same line).

- What was the incubation time, temperature, and CFU used for the study? How did the authors avoid the cross contamination during culture growth and incubation? 

All K. phaffii cultivations were performed according standard microbiology laboratory aseptic procedures, as reported in our previous paper (ref. 25). We chose not to insert the relevant protocol in the present paper as this coud be detected as plagiarism.

- There may be chances of cell death due to lack of sufficient humidity required for the culture growth. How did you maintain? 

As reported in ref 25, the yeasts were grown in shake flask liquid culture. For this reason, issues related to humidity do not apply.

- In the sentence “the grids were dried with Whatman filter paper and analyzed.”, it is unavoidable handling of the sample for TEM analysis. I suggest to remove it.

Given the context of comment 3, we are not sure whether the Reviewer is talking about K. phaffii cells or vault nanoparticle. In any case, TEM analysis was carried out on pure vault, and grid drying is a standard treatment of samples to be subjected to this analytic procedure. So, we judge it more appropriate to leave this sentence in the text.

COMMENT 4: In section 4.6, TEM analysis was conducted to observe morphological assessment. The culture visualization after drying is a challenging task for real image scanning. How did the authors manage to avoid heat induced culture damage and structural changes?. On the grid, there were several microscopic pores to trap the loaded sample. How the preferential adsorption of small size culture did the author prevent?

As pointed out in the previous point, our TEM analysis was performed on the pure vault NP, not on cultured cells. So, issues related to possible culture damage and structural changes are not relevant to this analytical procedure.

COMMENT 4: In section 4.7, the authors analyzed size. Please mention the nature of sample (transparent or turbid). Did the author dilute the sample before analysis? If yes, please include the details.

Thank you for this remark. Actually, we realize that we omitted to specify these features, which is unquestionably important. So, we have inserted in Materials and Methods (Section 4.7) the following sentence (lines 439-441):
Before analysis, the samples were spun down at 20,000 x g for 20 min, so they were cleared of any turbidity. Then, they were allowed to equilibrate for 2 min prior to measurement. Routinely, sample concentration was in the range 0.2-0.5 mg/mL. 

COMMENT 5: For the sentence “the mammary carcinoma cell lines SKBR3 535and the triple negative MDA-MB 231 Luc+, purchased from ATCC. SKBR3 cells were cul- 536tured in 50% Dulbecco’s Modified Eagle’s Medium High Glucose (DMEM-HG) and 50% 537HAM’S F12. MDA-MB 231 Luc + cells were cultured in 100% DMEM-HG with 1% of non- 538essential amino acids solution (MEM NEAA, 100X).” I recommend to shift this in material section. I found repetition of various materials with source in the text body.

Very respectfully, please note that the sentence mentioned by the Reviewer is already in the section Materials and Methods (Section 4.11), lines 525-530.

COMMENT 6: I suggest to replace figure 3 with fine line of curve setting. Figure 4 is not clear. I recommend to provide high resolution image of figure 4. 

Regarding Figure 3, we are sorry we cannot replace it with another with thinner lines. This is because in the figure we just present the output of the instrument, so to comply with this recommendation we should redraw the profiles by hand, which would be very unpractical if ever feasible.
Instead, regarding Figure 4, we now present a better resolution image (1000 dpi). In any case, the information it conveys is (and was) unambiguous.

COMMENT 7: trade mark is copyright related to owner of the material or tool. I suggest to remove the superscript of “TM” as present in “PageRuler™”.

7. We have deleted TM (lines 205 & 418).

COMMENT 8: In figure 5, standard error values are exceptionally higher than the mean. Please check for the value. It must not be too high.  

We present a new version of the figure, wherein mean values and standard deviations has been recalculated on the basis of a broader data set consisting of four replicates. This decreases to some extent the measurement scatter. On the other hand, we would like to remark that, if taken as a whole, densitometry and MS data corroborate each other so that our conclusions are definitely supported by the overall experimental picture.

COMMENT 9: I found the plagiarism report at 51%. It cannot be accepted for publication in this form.

As already explained to the Section Managing Editor, Mr. Wang, this apparent plagiarism is due to partial overlapping with the PhD thesis of Dr. Tomaino, who is also the first author of the manuscript.
Also, following my contacts with the competent officials of my University, I have learnt that the copyright of PhD theses is not detained by the University itself but entirely by the respective authors, in this specific case Dr. Tomaino herself. This disproves any allegation of plagiarism, as also Dr. Wang has agreed.

Round 2

Reviewer 2 Report

Comments and Suggestions for Authors

Recommended for publication